# Effects of Chronic LY341495 on Hippocampal mTORC1 Signaling in Mice with Chronic Unpredictable Stress-Induced Depression

**DOI:** 10.3390/ijms23126416

**Published:** 2022-06-08

**Authors:** Mi Kyoung Seo, Jung An Lee, Sehoon Jeong, Dae-Hyun Seog, Jung Goo Lee, Sung Woo Park

**Affiliations:** 1Paik Institute for Clinical Research, Inje University, Busan 47392, Korea; banaba66@inje.ac.kr (M.K.S.); jeongsh@inje.ac.kr (S.J.); 2Department of Psychiatry, Inje University Haeundae Paik Hospital, College of Medicine, Inje University, Busan 48108, Korea; bluesilver27@naver.com; 3Department of Healthcare Information Technology, Inje University, Gimhae 50834, Korea; 4Institute for Digital Antiaging and Healthcare, Inje University, Gimhae 50834, Korea; 5Department of Biochemistry, College of Medicine, Inje University, Busan 47392, Korea; daehyun@inje.ac.kr; 6Dementia and Neurodegenerative Disease Research Center, College of Medicine, Inje University, Busan 47392, Korea; 7Department of Convergence Biomedical Science, College of Medicine, Inje University, Busan 47392, Korea

**Keywords:** chronic unpredictable stress, depression, hippocampus, LY341495, mTORC1

## Abstract

In several rodent models, acute administration of the metabotropic glutamate 2/3 (mGlu2/3) receptor antagonist LY341495 induced antidepressant-like effects via a mechanism of action similar to that of ketamine. However, the effects of chronic mGlu2/3 antagonism have not yet been explored. Therefore, we investigated the effects of chronic LY341495 treatment on the mechanistic target of rapamycin complex 1 (mTORC1) signaling and the levels of synaptic proteins in mice subjected to chronic unpredictable stress (CUS). LY341495 (1 mg/kg) was administered daily for 4 weeks to mice with and without CUS exposure. After the final treatment, the forced swimming test (FST) was used to assess antidepressant-like effects. The hippocampal levels of mTORC1-related proteins were derived by Western blotting. Chronic LY341495 treatment reversed the CUS-induced behavioral effects of FST. CUS significantly reduced the phosphorylation of mTORC1 and downstream effectors [eukaryotic translation initiation factor 4E-binding protein 1 (4E-BP-1) and small ribosomal protein 6 (S6)], as well as the expression of synaptic proteins postsynaptic density-95 (PSD-95) and AMPA receptor subunit GluR1 (GluA1) in the hippocampus. However, chronic LY341495 treatment rescued these deficits. Our results suggest that the activation of hippocampal mTORC1 signaling is related to the antidepressant effect of chronic LY341495 treatment in an animal model of CUS-induced depression.

## 1. Introduction

Accumulating evidence indicates that the modulation of the glutamatergic system might be useful for treating depression. Many studies have shown that single injections of ketamine [a noncompetitive N-methyl-D-aspartate (NMDA) receptor antagonist] exert rapid and sustained antidepressant effects in patients with major depressive disorder and treatment-resistance depression [1,2,3]. The molecular mechanisms have been investigated in many animal studies. Activation of mechanistic target of rapamycin complex 1 (mTORC1) signaling by α-amino-3-hydroxy-5-methylisoxazole-4-propionate (AMPA) receptor activation may play a crucial role. Ketamine increases extracellular glutamate release, followed by AMPA receptor activation, and, in turn, the release of brain-derived neurotrophic factor (BDNF) and the stimulation of the tropomyosin-related kinase B (TrkB) receptor. The phosphatidyl inositol-3 kinase (PI3K)-Akt (protein kinase B) and Ras-mitogen-activated protein kinase (MAPK) pathways are then activated, in turn triggering the mTORC1 signaling that enhances messenger ribonucleic acid (mRNA) translation via the downstream substrates small ribosomal protein 6 (S6) kinase and eukaryotic translation initiation factor 4E-binding protein 1 (4E-BP-1). This increases the expression levels of the synaptic proteins’ postsynaptic density-95 (PSD-95) and the AMPA receptor AMPA receptor subunit GluR1 (GluA1) subunit; synaptic plasticity is thus enhanced [4,5].

Although ketamine has robust antidepressant effects, there are numerous side effects, including psychotomimetic and dissociative symptoms, as well as abuse potential, which limit general prescription. Drugs that target the glutamatergic system and exert antidepressant effects similar to those of ketamine, but without the side effects, are urgently needed. The metabotropic glutamate 2/3 (mGlu2/3) receptors are G-protein-coupled receptors that negatively regulate adenylyl cyclase activity and decrease glutamate and monoamine release. Antagonists of such receptors (e.g., LY341495 and MGS0039) enhance glutamate transmission in brain regions which are known to be affected in depression [6]. Many studies have shown that LY341495 and MGS0039 share some of the antidepressant actions of ketamine. LY341495 and MGS0039 provided rapid and prolonged antidepressant-like effects in the forced swimming test (FST) and tail suspension test (TST) [7,8,9]; these effects were associated with AMPA receptor stimulation, activation of mTORC1 signaling, and increased levels of synaptic proteins in the prefrontal cortex (PFC) [10,11]. We previously showed that both LY341495 and ketamine enhanced neural plasticity by activating the AMPA receptor and mTORC1 signaling in rat hippocampal neurons [12]. Although the antidepressant actions of mGlu2/3 antagonists have previously been described, most studies were performed in animals not previously exposed to depression-inducing stressors. Moreover, most studies evaluated the antidepressant-like effects in the FST or TST 24 h after acute drug administration. To the best of our knowledge, only three studies have explored the antidepressant-like effects of mGlu2/3 receptor antagonists on stress-induced depression. In one study, a single injection of LY341495 reversed the anhedonia caused by chronic unpredictable stress (CUS) within 48 h; the effect persisted for 10 days [13]. Another study revealed antidepressant-like effects of LY341495, from a single and sub-chronic (3 days) administration, in a CUS model. The effects were associated with the activation of the mTORC1 signaling pathway [14]. Finally, a study using a social defeat stress model reported antidepressant-like effects and enhanced synaptogenesis after a single injection of MGS0039 [15]. Furthermore, acute ketamine administration improves synaptic plasticity in rodents. On the other hands, chronic ketamine administration reduced the expression of synaptic proteins GluA1, GluA2, synaptophysin, and PSD-95 and spine density, and induced synaptic transmission and plasticity impairments in the hippocampus [16,17,18]. However, no study has investigated the effects of chronic mGlu2/3 antagonist treatment on mTORC1 signaling and synaptic protein expression in a stressful environment. We aimed to determine whether chronic LY341495 administration caused long-lasting changes in mTORC1 signaling. Therefore, we established a mouse model of CUS-induced depression, and analyzed the effects of chronically administered LY341495 on hippocampal mTORC1 signaling.

## 2. Results

### 2.1. Behavioral Effect of Chronic LY341495 Administration in a Mouse Model of CUS-Induced Depression

To determine if CUS induces stress, we monitored body weight (Figure 1A). Two-way ANOVA revealed significant effects of CUS as a main factor during all weeks, but no LY341495 and CUS × LY341495 interaction: week 1 (CUS, F_(1,27)_ = 72.400, *p* < 0.001; LY341495, F_(1,27)_ = 2.260, *p* = 0.145; CUS × LY341495, F_(1,27)_ = 0.251, *p* = 0.621); week 2 (CUS, F_(1,27)_ = 75.400, *p* < 0.001; LY341495, F_(1,27)_ = 1.110, *p* = 0.320; CUS × LY341495, F_(1,27)_ = 0.247, *p* = 0.623); week 3 (CUS, F_(1,27)_ = 45.400, *p* < 0.001; LY341495, F_(1,27)_ = 0.093, *p* = 0.765; CUS × LY341495, F_(1,27)_ = 0.301, *p* = 0.588); week 4 (CUS, F_(1,26)_ = 29.400, *p* < 0.001; LY341495, F_(1,26)_ = 1.180, P = 0.288; CUS × LY341495, F_(1,26)_ = 0.924, *p* = 0.345). In the post hoc test, CUS significantly decreased mice body weight at week 1 (*p* < 0.001), week 2 (*p* < 0.001), week 3 (*p* < 0.001), and week 4 (*p* = 0.015), compared to the control group.

The antidepressant-like effect of chronic LY341495 (1 mg/kg, i.p.) was tested using the FST (Figure 1B). Two-way ANOVA revealed significant main effects of CUS [F_(1,26)_ = 10.150, *p* = 0.004] and LY341495 [F_(1,26)_ = 9.968, *p* = 0.004], but no CUS × LY341495 interaction effect [F_(1,26)_ = 2.768, *p* = 0.108]. In the post hoc test, CUS significantly increased the immobility time compared to that of non-stressed mice (*p* = 0.008 vs. the control group). Chronic LY341495 administration significantly reduced the immobility time (*p* = 0.017 vs. the CUS group).

### 2.2. Effects of Chronic LY341495 Treatment on Hippocampal mTORC1 Signaling in CUS-Exposed Mice

The hippocampal levels of phosphorylated mTORC1, 4E-BP-1 and S6 after the behavioral test were determined by Western blotting. CUS exposure decreased the level of phospho-mTORC1 (*p* = 0.012 vs. the control group); LY341495 treatment enhanced the hippocampal phospho-mTORC1 level in CUS-exposed mice (*p* = 0.036 vs. the CUS group, Figure 2A). Two-way ANOVA indicated a significant CUS × LY341495 interaction effect [F_(1,20)_ = 8.825, *p* = 0.008], but there was no main effect of either CUS [F_(1,20)_ = 3.369, *p* = 0.081] or LY341495 [F_(1,20)_ = 2.405, *p* = 0.137]. CUS downregulated the hippocampal phospho-4E-BP-1 level (*p* = 0.001 vs. the control group, Figure 2B) and the phospho-S6 level (*p* = 0.001 vs. the control group, Figure 2C); both are downstream factors of mTORC1. These effects were reversed by chronic LY341495 treatment (phospho-4E-BP-1: *p* = 0.032 vs. the CUS group; phospho-S6: *p* = 0.020 vs. the CUS group). In terms of the phospho-4E-BP-1 level, two-way ANOVA revealed a significant interaction between CUS and LY341495 [F_(1,22)_ = 7.756, *p* = 0.011], a significant main effect of CUS [F_(1,22)_ = 10.400, *p* = 0.004], and a trend toward a main effect of LY341495 [F_(1,22)_ = 4.006, *p* = 0.058]. In terms of the phospho-S6 level, two-way ANOVA revealed significant main effects of CUS [F_(1,25)_ = 17.500, *p* < 0.001] and LY341495 [F_(1,25)_ = 7.901, *p* = 0.009], but no interaction effect [F_(1,25)_ = 2.971, *p* = 0.097].

Figure 3A shows that CUS did not affect the phospho-Akt level (*p* > 0.999 vs. the control group), but chronic LY341495 treatment significantly increased this level in CUS-exposed mice (*p* = 0.010 vs. the CUS group). Two-way ANOVA revealed a significant interaction between CUS and LY341494 [F_(1,25)_ = 6.256, *p* = 0.019], a trend toward a main effect of CUS [F_(1,25)_ = 3.996, *p* = 0.057], and a significant main effect of LY341495 [F_(1,25)_ = 7.855, *p* = 0.010].

mTORC1 regulates the protein synthesis required for synaptogenesis. Thus, we examined the hippocampal expression of PSD-95 and GluA1. CUS exposure significantly decreased the levels of PSD-95 (*p* = 0.013 vs. the control group, Figure 3B) and GluA1 (*p* = 0.048 vs. the control group, Figure 3C). Chronic administration of LY341495 completely reversed these CUS-induced decreases (PSD-95: *p* = 0.007 vs. the CUS group; GluA1: *p* = 0.018 vs. the CUS group). Two-way ANOVA indicated a significant interaction between CUS and LY341495, but indicated no main effect of either CUS or LY341495 on the PSD-95 level [CUS × LY341495: F_(1,26)_ = 19.590, *p* < 0.001; CUS: F_(1,26)_ = 0.033, *p* = 0.857; LY341495: F_(1,26)_ = 0.787, *p* = 0.383] or GluA1 level [CUS × LY341495: F_(1,22)_ = 13.020, *p* = 0.001; CUS: F_(1,23)_ = 0.098, *p* = 0.757; LY341495: F_(1,22)_ = 1.744, *p* = 0.200].

## 3. Discussion

In the present study, the FST immobility induced by 4 weeks of CUS was significantly reduced by chronic LY341495 treatment, which also attenuated the CUS-induced decreases in hippocampal mTORC1 signaling. Thus, the antidepressant-like effects of chronic LY341495 may be associated with activation of mTORC1 signaling.

The CUS protocol mimics everyday stress; it is used to study the stress response and consistently induces behavioral alterations typical of chronic stress [19]. For the study, three weeks of CUS exposure reduced the sucrose preference test score (indicating anhedonia), and increased the feeding latency in a novel environment, indicating increased anxiety [20]. In addition, eight weeks of CUS administration induced depression-like behaviors in the FST, TST, and sucrose preference tests [21,22]. CUS affected brain mTORC1 phosphorylation levels. Overall, eight weeks of stress decreased mTORC1, p70S6 kinase (p70S6K), and 4E-BP-1 phosphorylation, as well as PSD-95 expression, in the hippocampus and amygdala, but not in the frontal cortex or hypothalamus [21]. In another study, eight weeks of CUS reduced mTORC1, p70S6K, and 4E-BP-1 phosphorylation in the hippocampus and PFC [22], and downregulated the phosphorylation of hippocampal Akt and extracellular signal-regulated kinase (ERK) (two upstream regulators of mTORC1), as well as hippocampal PSD-95 and synapsin I expression [22]. In mice which were stressed for eight weeks, the phosphorylated forms of mTORC1, p70S6K, S6, 4E-BP1, ERK, Akt, and GluA1 were decreased in the amygdala, but not in the frontal cortex, hippocampus, or dorsal raphe [23]. Stress for three weeks decreased the levels of the synaptic proteins PSD-95, synapsin I, and GluA1 in the PFC, but did not significantly affect the levels of phosphorylated mTORC1, p70S6K, or 4E-BP-1 [20]. Thus, the effects of CUS on mTORC1 signaling may be specific to various brain regions, and could depend on CUS duration.

Akt and ERK phosphorylation is associated with depression and antidepressant actions in humans and rodents. This phosphorylation plays important roles in transcriptional and translational activation via mTORC1 signaling, which enhances neural plasticity [4]. In mice, chronic stress decreases hippocampal Akt and ERK phosphorylation [24,25,26], in turn impairing hippocampal neural plasticity by downregulating mTORC1 signaling; in turn, this induces depression-like behaviors. However, we found no significant change in hippocampal Akt phosphorylation status in mice exposed to four weeks of CUS. On the other hand, LY341495 markedly increased the phosphorylated Akt level after CUS. There is no obvious explanation for this, and further analysis of ERK phosphorylation is needed. Our CUS protocol may downregulate mTORC1 signaling by inhibiting ERK.

Similar to ketamine, mGlu2/3 receptor antagonists exhibit fast-onset antidepressant effects. Western blotting indicated that chronic LY341495 treatment increased the phosphorylation of hippocampal Akt (an upstream regulator of mTORC1) in CUS mice. mGlu2/3 receptor antagonists mainly modulate glutamate release by blocking the mGlu2/3 autoreceptor [27]. LY341495 indirectly stimulates the AMPA receptor signaling via the postsynaptic actions of glutamate, thereby increasing activity-dependent BDNF release and BDNF synthesis, followed by TrkB receptor stimulation and PI3K/Akt (ERK)/mTORC1 downstream signaling (Figure 4). This signaling ultimately increases the translation of transcripts encoding the synaptic proteins PSD-95 and the AMPA receptor GluA1 subunit. LY34495 exerts antidepressant effects by increasing the synthesis of these synaptic proteins via stimulation of the AMPA receptor, BDNF/TrkB/PI3K/Akt (ERK), and through mTORC1 signaling [10,12,28]. This study showed that chronic LY341495 treatment reversed chronic stress-induced decreases in hippocampal synaptic protein levels. Previously, we reported that LY341495 attenuated the dexamethasone-induced decreases in dendritic outgrowth and spinal density by stimulating AMPA receptor-mTORC1 signaling in primary hippocampal cultures [12]. Together, the data suggest that restoration of hippocampal synaptic functions disrupted by stress may explain the antidepressant effects of LY341495.

We have observed the increases of hippocampal mTORC1 signaling after chronic LY341495 administration in rats subjected to CUS. Previous studies have shown that the antidepressant effects of ketamine in behavioral tests are associated with AMPA receptor-mediated upregulation of BDNF and mTORC1 signaling in rat hippocampus and PFC [5,29,30,31]. These two areas are the limbic brain regions directly involved in depression and antidepressant actions [32,33]. Thus, studies on the mechanism of LY341495 exerting antidepressant effects has also been investigated in PFC or mixed tissue composed of the PFC and the hippocampus. Previously, we reported an in vitro study showing that LY341495 promoted hippocampal dendritic outgrowth and synaptic protein expression via stimulating mTORC1 signaling [12]. Since it was necessary to perform an in vivo study, we investigated the mechanism in the hippocampus.

The antidepressant effects of LY341495 may involve serotonergic transmission (Figure 4) [34]. Karasawa et al. reported that mGlu2/3 receptor antagonists increased 5-HT neuronal firing in the dorsal raphe nucleus (DRN) and the extracellular serotonin (5-HT) level of the medial Prefrontal Cortex (mPFC) by activating postsynaptic AMPA receptors [35,36]; this was subsequently confirmed [37,38,39,40,41,42]. The antidepressant effects of LY341495 and ketamine on the novelty-suppressed feeding test and FST scores were blocked by the depletion of 5-HT [i.e., by pretreatment with para-chlorophenylalanine methyl ester (PCPA), an inhibitor of 5-HT synthesis] [37,38,39,40]. A microinjection of LY341495 into the mPFC increased the number of 5-HT neurons in the DRN, but this effect was blocked by microinjection of the AMPA receptor antagonist 2,3-dioxo-6-nitro-7-sulfamoyl-benzo[f]quinoxaline (NBQX) [39]. Thus, it is conceivable that LY341495 stimulates mPFC AMPA receptors, thereby activating the 5-HT neurons of the DRN, and ultimately increasing 5-HT release into the mPFC. Fukumoto et al. reported an interaction between the serotoninergic system and synaptic plasticity [41]. Importantly, the antidepressant effect of LY341495 (revealed by the FST) was blocked by a microinjection of a serotonin 1A (5-HT_1A)_ receptor antagonist (WAY100635) into the mPFC; a 5-HT_1A_ receptor agonist showed no such effect. Furthermore, the antidepressant effect of LY341495 was blocked by an injection of the PI3K inhibitor LY294002 and mTORC1 inhibitor rapamycin into the mPFC [41]. LY341495 increased mPFC Akt phosphorylation, but this effect was blocked by an injection of a 5-HT_1A_ receptor antagonist and PI3K inhibitor. These findings suggest that LY341495 activates PI3K/Akt/mTORC1 signaling downstream of the 5-HT_1A_ receptor by stimulating that receptor in the mPFC, and exerts antidepressant effects by improving synaptic plasticity [41].

There are several limitations to the current study. Firstly, we performed the FST to assess the depression-like phenotype. Many studies have applied this model as a behavioral test, but using the FST alone is not sufficient. The FST is a more appropriate tool for screening potential antidepressants. Therefore, additional behavioral tests, such as the sucrose preference test, are needed to measure anhedonia, a major symptom of depression. Secondly, we investigated alteration in mTORC1 signaling only in the hippocampus. Further analyses are needed in the PFC, amygdala, and dorsal raphe. Finally, additional signaling components, such as PI3K, ERK, p70S6K, and TrkB receptor, as shown in Figure 4, should be analyzed. In summary, we found that chronic LY341495 treatment modulated the activation of hippocampal mTORC1 signaling under stressful, but not normal, conditions. Chronic LY341495 treatment attenuated the CUS-induced reduction in hippocampal phosphorylation of mTORC1, and its downstream regulators 4E-BP1 and S6. Chronic LY341495 treatment attenuated the CUS-induced reduction in hippocampal PSD-95 and GluA1 expression. To further explore the “mTORC1 signaling hypothesis”, which involves the glutamatergic and 5-HTergic systems, it is important to establish whether the antidepressant action of LY341495 is abolished by the hippocampal infusion of AMPA and 5-HT_1A_ receptor antagonists, as well as PI3K and mTORC1 inhibitors, in mice exposed to chronic stress.

## 4. Materials and Methods

### 4.1. Animals

Male 7-week-old C57BL/6J mice (Orient Bio, Seongnam-si, Gyeonggi-do, Korea) were employed. Before use, all mice were maintained under standard laboratory conditions (temperature, 23 ± 1 °C; 12/12 h light/dark cycle, lights on at 07:00; humidity, 55 ± 10%) for 1 week with free access to food and water. All animal experiments were conducted in accordance with the guidelines of the Institutional Animal Care and Use Committee (IACUC) of Inje University, Republic of Korea, and were approved by the IACUC of the College of Medicine of Inje University (approval no. 2016–044).

### 4.2. Drug Administration and Experimental Groups

LY341495 was purchased from Tocris Bioscience (Bristol, Avon, UK), dissolved in 0.9% (*w*/*v*) saline (vehicle), and intraperitoneally injected (1 mg/kg/day) for 4 weeks. This dose has been reported to exert antidepressant-like effects in the FST and TST [9,11,41]. LY341495 was administered daily between 09:00 and 10:00, 1 h prior to stress treatment for 4 weeks. Each group consisted of 6–8 mice. Mice were randomly assigned to one of the following groups: control group (non-stressed mice that received the vehicle), LY341495 group (non-stressed mice that received LY341495), CUS group (stressed mice that received the vehicle), and CUS + LY341495 group (stressed mice that received LY341495).

### 4.3. Chronic Unpredictable Stress (CUS) Induction

A previously described CUS procedure [42] was modified for this study. The following stressors were used: an empty cage (24 h), restraint stress (4 h), 45 °C cage tilt (4 h), 4 °C cold swimming (5 min), nipping 1 cm from the end of the tail (1 min), food and water deprivation (24 h), and wet bedding (100 mL of water in sawdust) (24 h). One stressor was applied daily. The stressors were administered randomly over a 1-week period every day for a total of 4 weeks. Body weight was monitored once a week.

### 4.4. Forced Swimming Test (FST)

Control, LY31495, CUS, and CUS-plus-LY341495 mice (*n* = 6–8) were subjected to the FST, as described previously, with minor modifications [43]. The FST is useful for screening compounds with the potential to treat depression. We assessed the effect of LY341495 in terms of the immobility time. After a period of 24 h after the last CUS protocol, mice were subjected to the FST. Briefly, the mice were placed in transparent plastic cylinders (10 cm in diameter, 25 cm in height) filled with water (23–25 °C) for 7 min. All behaviors were videotaped, and subsequently scored. After the first 2 min, the immobility time was measured over 5 min by a trained experimenter blinded to the group assignment. Immobility was defined as passive floating (without struggling).

### 4.5. Western Blotting

After the FST, the mice were rapidly sacrificed by cervical dislocation. The hippocampus was dissected, frozen in liquid nitrogen, and stored at −80 °C. Western blotting was performed as described previously [12]. Briefly, tissue samples were homogenized in ice-cold NP-40 buffer (Elpis Biotech, Lexington, MA, USA) with the cOmplete 1× protease inhibitor cocktail (Roche, Mannheim, Baden-Wurttemberg, Germany). The homogenate was centrifuged (1000× *g*, 15 min, 4 °C), and the supernatant was collected. Protein concentrations were measured using the Bradford assay. For immunoblotting at 4 °C overnight, the following primary antibodies were individually added to Tris-buffered saline with Tween-20 (TBS-T): anti-phospho-mTORC1 (Ser2448, #2971), anti-mTORC1 (#2972), anti-phospho-Akt (Ser473, #9271), anti-Akt (#9272), anti-phosho-4E-BP-1 (Thr37/46, #2855), anti-4E-BP-1 (#9452), anti-phospho-S6 (Ser240/244, #2215), anti-S6 (#2217), anti-PSD-95 (1:1000, #3450), and anti-GluA1 (1:1000, #13185; all from Cell Signaling Technology, Danvers, MA, USA); and anti-α-tubulin (1:2000; T9026, Sigma, St. Louis, MO, USA). On the following day, the membranes were washed three times in TBS-T for 10 min, and incubated for 1 h in TBS-T with horseradish peroxidase-conjugated secondary antibodies at room temperature: goat-anti-rabbit IgG (1:2000, sc-2004; Santa Cruz Biotechnology, Santa Cruz, CA, USA) for anti-phospho-mTORC1, anti-mTORC1, anti-phospho-Akt, anti-Akt, anti-phospho-4E-BP-1, anti-4E-BP-1, anti-phospho-S6, anti-S6, anti-PSD-95, and anti-GluA1; and anti-mouse IgG (1:10,000, A4416; Sigma) for anti-α-tubulin.

### 4.6. Statistical Analysis

All data are expressed as mean ± standard error of the mean (SEM). Statistical analyses were performed with GraphPad Prism 9.3.1. (GraphPad Software Inc., La Jolla, CA, USA). The main and interaction effects of CUS and LY341495 in the behavioral tests and Western blotting data were determined by two-way ANOVA. Tukey’s multiple-comparison test was used for post hoc comparisons. *p*-values < 0.05 were considered to indicate statistically significant differences.

## Figures and Tables

**Figure 1 ijms-23-06416-f001:**
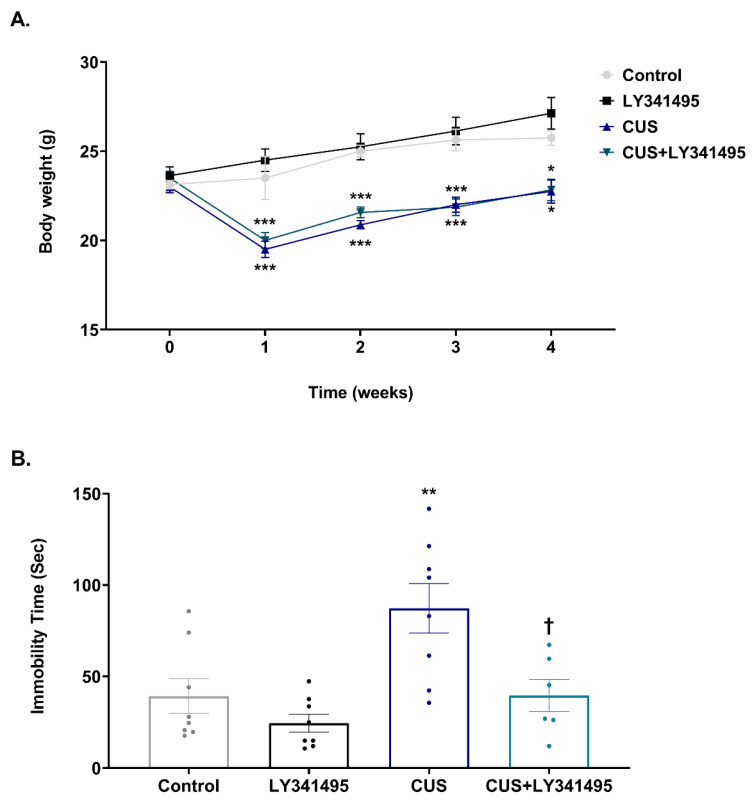
Antidepressant-like effects of chronic LY341495 administration in chronic unpredictable stress (CUS)-exposed mice, as indicated by the forced swimming test (FST). Mice were treated daily with LY341495 (1 mg/kg) or vehicle (1 mL/kg, 0.9% saline) with (the CUS and CUS + LY341495 groups) or without (the control and LY341495 groups) CUS for 4 weeks. Body weight (**A**) was measured every week for 4 weeks. The FST immobility time (**B**) was measured 24 h after the last CUS session. The control (non-stressed mice that received the vehicle), LY341495 (non-stressed mice that received LY341495), CUS (stressed mice that received the vehicle), and CUS + LY341495 (stressed mice that received LY341495) groups were compared. Values are expressed as mean ± SEM. * *p* < 0.05 vs. the control group, ** *p* < 0.01 vs. the control group, *** *p* < 0.001 vs. the control group; ^†^
*p* < 0.05 vs. the CUS group.

**Figure 2 ijms-23-06416-f002:**
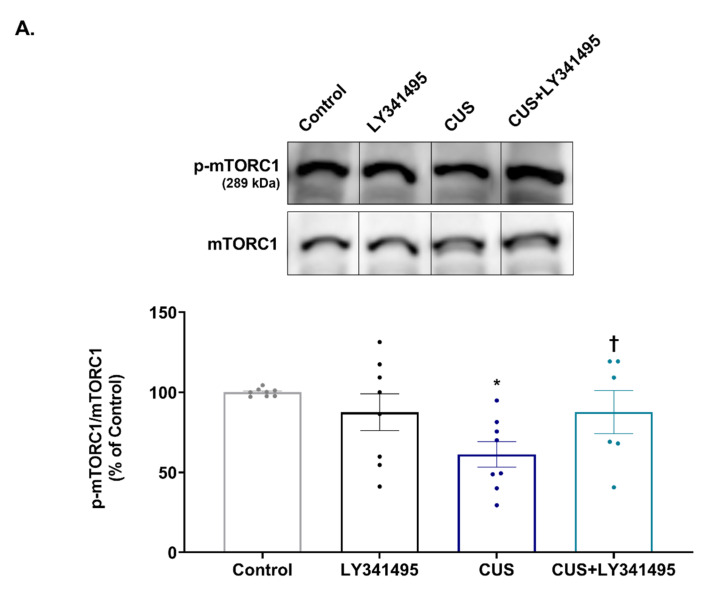
Effects of chronic LY341495 administration on the hippocampal levels of phosphorylated mechanistic target of rapamycin complex 1 (mTORC1), eukaryotic translation initiation factor 4E-binding protein 1 (4E-BP-1), and small ribosomal protein 6 (S6) in chronic unpredictable stress (CUS)-exposed mice. Mice were treated daily with LY341495 (1 mg/kg) or vehicle (1 mL/kg, 0.9% saline) with (the CUS and CUS + LY341495 groups) or without (the control and LY341495 groups) CUS for 4 weeks. The hippocampal levels of phospho-mTORC1 (**A**), phospho-4E-BP-1 (**B**), and phospho-S6 (**C**) were determined by Western blotting, and normalized to the total protein levels. The original images from which we constructed Figure 2A–C are shown as Appendix A. Values represent means ± SEM expressed as percentages of those of the control group. * *p* < 0.05 vs. the control group, ** *p* < 0.01 vs. the control group; ^†^
*p* < 0.05 vs. the CUS group.

**Figure 3 ijms-23-06416-f003:**
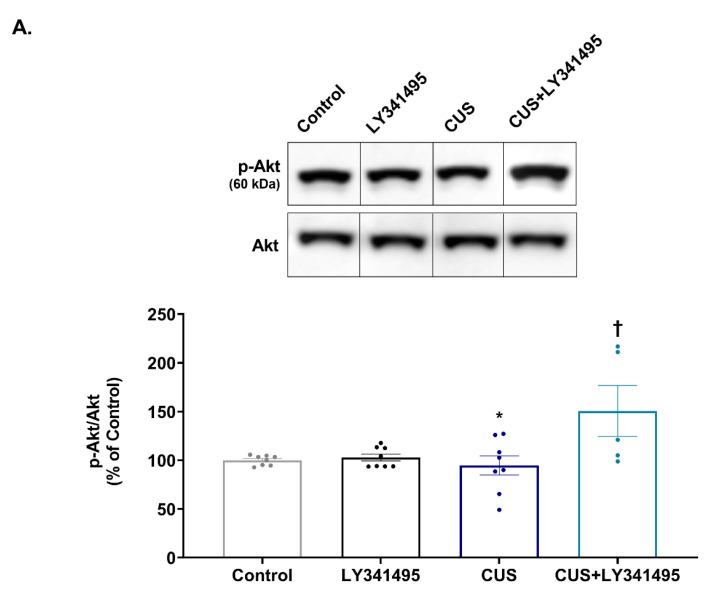
Effects of chronic LY341495 administration on protein kinase B (Akt) phosphorylation and postsynaptic density-95 (PSD-95) and AMPA receptor subunit GluR1 (GluA1) hippocampal expression in chronic unpredictable stress (CUS)-exposed mice. Mice were treated daily with LY341495 (1 mg/kg) or vehicle (1 mL/kg, 0.9% saline) with (the CUS and CUS + LY341495 groups) or without (the control and LY341495 groups) CUS for 4 weeks. The hippocampal levels of phospho-Akt (**A**), PSD-95 (**B**), and GluA1 (**C**) were determined by Western blotting. The levels of phosphorylated and synaptic proteins were normalized to the total protein and α-tubulin levels, respectively. The original images from which we constructed Figure 3A–C are shown as Appendix A. Values represent mean ± SEM expressed as percentages of those of the control group; * *p* < 0.05 vs. the control group; ^†^
*p* < 0.05 vs. the CUS group, ^††^
*p* < 0.01 vs. the CUS group.

**Figure 4 ijms-23-06416-f004:**
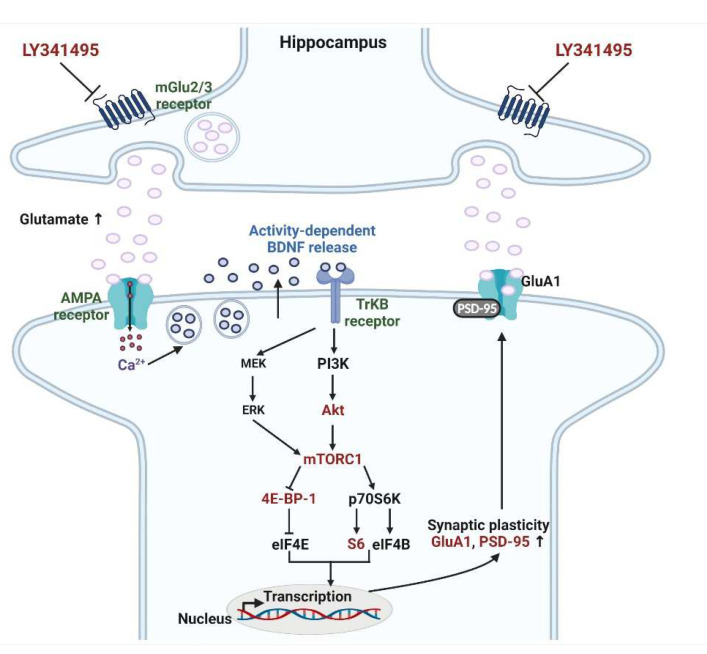
Putative signaling pathway underlying the antidepressant actions of LY341495 in the hippocampus. LY341495 increases extracellular glutamate release, thereby inducing BDNF release. BDNF stimulates the TrkB receptor and downstream signaling pathways PI3K/Akt and MAPK, in turn activating mTORC1 signaling. mTORC1 activates p70S6K and inhibits 4E-BP-1, thereby increasing the translation of transcripts encoding the synaptic proteins PSD-95 and GluA1 (which enhance AMPA receptor transmission). The synaptic plasticity is thus enhanced, and, overall, this process is associated with the antidepressant effects of LY341495. The molecular pathway marked in red are the pathway observed in the current study. Akt, protein kinase B; AMPA, α-amino-3-hydroxy-5-methylisoxazole-4-propionic acid; BDNF, brain-derived neurotrophic factor; Ca^2+^, calcium; eIF4E, eukaryotic translation initiation factor 4E; eIF4B, eukaryotic translation initiation factor 4B; ERK, extracellular signal-regulated kinase; GluA1, AMPA receptor subunit GluA1; MEK, MAP/ERK kinase; mGlu2/3, metabotropic glutamate 2/3; mTORC1, mechanistic target of rapamycin complex 1; p70S6K, p70S6 kinase; PI3K, phosphatidyl inositol-3 kinase; PSD-95, postsynaptic density-95; S6, small ribosomal protein 6; TrKB, tropomyosin-related kinase B; 4E-BP-1, eukaryotic translation initiation factor 4E (eIF4E)-binding protein 1. Original illustration created by MK Seo using BioRender (biorender.com, accessed on 15 May 2022).

## Data Availability

The data presented in this study are available on request from the corresponding author.

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
