# Peer review of "Effects of Chronic LY341495 on Hippocampal mTORC1 Signaling in Mice with Chronic Unpredictable Stress-Induced Depression"

_ijms, 2022, doi:10.3390/ijms23126416_

Round 1

Reviewer 1 Report

Seo et al subjected mice to 4 weeks of chronic unpredictable stress (CUS) and to the mGlu2/3 receptor antagonist LY341495 (1 mg/kg i.p.). LY341495 rescued the CUS-induced increased immobility in forced swim test (FST). LY341495 also normalized the CUS-induced attenuation of mTOR signaling (as assessed by p-mTORC1, p-4E-BP1, p-S6) and of synaptic markers PSD95 and GluA1.

Major

1) Since the stated starting point of the study is that acute mGlu2/3 receptor antagonists have fast-onset antidepressant effects similar to ketamine, the rational of looking into effects of chronic LY341495 is unclear.

2) Also, why to focus on the hippocampus in the context of forced swim test? Figure 4 does not even mention the hippocampus, instead showing the mPFC and DRN, which were not studied here.

3) The conclusion at the end of the abstract may be too strong, since as the authors themselves acknowledge at the end of the discussion, the presented data is correlative, not causative.

In summary, the paper is straightforward and presented clearly, but its value is questionable.

Minor

4) I guess that the authors stretched the bands from the original gels for the actual figures, which does not make sense.

5) I guess the first paragraph after each figure is the legend, but it is unclear in the current format.

6) 4.2. Drug administration includes redundant information, for example 4 weeks and 28 days.

7) Line 287: “for a further” should be “for a total of”

8) Line 299: cervical dislocation is not anesthesia.

Author Response

Summary of changes made

We appreciate the Reviewers feedback on our manuscript. All suggested corrections have been made.

Reviewer: 1

< Major >

1) Since the stated starting point of the study is that acute mGlu2/3 receptor antagonists have fast-onset antidepressant effects similar to ketamine, the rational of looking into effects of chronic LY341495 is unclear.

→ Thank you for your comment. Previous studies showed that, in contrast to acute treatment, chronic ketamine treatment had negative effects on synaptic plasticity. Up to now, the effects of chronic LY341495 were not investigated. This issue was described with new references in Introduction section (line 80-84).

2) Also, why to focus on the hippocampus in the context of forced swim test? Figure 4 does not even mention the hippocampus, instead showing the mPFC and DRN, which were not studied here.

→ Thank you for your comment. To determine antidepressant effect of chronic LY341495 administration, the FST was performed. Upregulation of hippocampal mTORC1 signaling may be one of the mechanisms explaining chronic LY341495-associated antidepressant effect. Therefore, Figure 4 has been revised to putative signaling observed in the hippocampus. mPFC and DRN, which were not investigated in the current study, were removed.

3) The conclusion at the end of the abstract may be too strong, since as the authors themselves acknowledge at the end of the discussion, the presented data is correlative, not causative.

In summary, the paper is straightforward and presented clearly, but its value is questionable.

→ Thank you for your comment. The conclusion of the abstract was rewritten as follows.

< Our results suggest that the activation of hippocampal mTORC1 signaling is related to the anti-depressant effect of chronic LY341495 treatment in an animal model of CUS-induced depression. >

< Minor >

4) I guess that the authors stretched the bands from the original gels for the actual figures, which does not make sense.

→ Thank you for your comment. All bands were changed to those of the original gel without stretching.

5) I guess the first paragraph after each figure is the legend, but it is unclear in the current format.

→ Thank you for your comment. The first paragraph of each figure has be corrected as follows.

< Mice were treated daily with LY341495 (1 mg/kg) or vehicle (1 mL/kg, 0.9% saline) with (the CUS and CUS + LY341495 groups) or without (the control and LY341495 groups) CUS for 4 weeks. >

6) 4.2. Drug administration includes redundant information, for example 4 weeks and 28 days.

→ Thank you for your comment. It was corrected to “4 weeks”.

7) Line 287: “for a further” should be “for a total of”

→ Thank you for your comment. Revised as suggested.

8) Line 299: cervical dislocation is not anesthesia.

→ Thank you for your comment. “anesthetized” has been changed to “sacrificed”.

Reviewer 2 Report

The authors reported that depression-like behaviour induced by chronic unpredictable stress (CUS) in mice was reduced by repeated mGlu2/3 receptor antagonist LY341495 treatment. Moreover, Western blot experiments revealed that the chronic LY341495 treatment reversed the CUS-induced changes in hippocampal phosphorylation of mTORC1, and its downstream regulators 4E-BP1 and S6, as well as in the the level of  Akt phosphorylation, PSD-95 and GluA 1. It is indicated that the mechanism of  LY341495 antidepressant-like activity involves modulation of hippocampal mTORC1 signaling under stressful but not normal conditions. This is a well performed study and the methods were properly chosen. The obtained data are quite original and interesting. On the other hand, this study has also some limitations, which should be pointed out in the Discussion.

Remarks:

1)      The FST is a reliable test for screening potential antidepressants. However,  it does not measure anhedonia, which is the key symptom of depression. Therefore, an additional test, e.g., the sucrose test would increase scientific value of this study. As the authors mentioned, the CUS was modified in the present study. Was it validated in an anhedonia test?

2)     In the Introduction the authors compared the effects of ketamine to those of  LY341495. Indeed, some studies have shown  functional synaptic changes in prefrontal cortext (PFC) after ketamine administration, which correlated with the antidepressant effects of the drug in animal models of depression (Koike et al. 2011). Thus, a rationale should be given why in the present study the biochemical changes were measured in the hippocampus instead of the PFC?

3)      The proposed “mTORC1 signaling hypothesis” shown in Fig. 4 was solely based on correlation of  behavioural test (FST) results with some biochemical changes. No experiments were performed in order to find out, if there is a functional link between these phenomena.

4)      This paper needs a language revision. Some sentences are not precise enough, e.g., see the Abstract: “Therefore, we investigated the effects of chronic LY341495 treatment on signaling by mechanistic target of rapamycin complex 1 (mTORC1) and the levels of synaptic proteins in mice subjected to chronic unpredictable stress (CUS) “.

Author Response

Summary of changes made

We appreciate the Reviewers feedback on our manuscript. All suggested corrections have been made.

Reviewer: 2

1) The FST is a reliable test for screening potential antidepressants. However, it does not measure anhedonia, which is the key symptom of depression. Therefore, an additional test, e.g., the sucrose test would increase scientific value of this study. As the authors mentioned, the CUS was modified in the present study. Was it validated in an anhedonia test?

→ We agree with your opinion. It is not sufficient to use only FST as the outcome measure. This is a limitation of this study. This issue was described in Discussion section as follows.

< There are several limitations to the current study. Firstly, we preformed the FST to assess the depression-like phenotype. Many studies have applied this model as a behavioral test, but using the FST alone is not sufficient. The FST is a more appropriate toll for screening potential antidepressants. Therefore, additional behavioral tests such as the sucrose preference test are needed to measure anhedonia, a major symptom of depression. >

2) In the Introduction the authors compared the effects of ketamine to those of LY341495. Indeed, some studies have shown functional synaptic changes in prefrontal cortext (PFC) after ketamine administration, which correlated with the antidepressant effects of the drug in animal models of depression (Koike et al. 2011). Thus, a rationale should be given why in the present study the biochemical changes were measured in the hippocampus instead of the PFC?

→ Thank you for your comment. This issue was described with new references in Discussion section as follows.

< We have observed the increases of hippocampal mTORC1 signaling after chronic LY341495 administration in rats subjected to CUS. Previous studies have shown that the antidepressant effects of ketamine in behavioral tests are associated with AMPA receptor-mediated upregulation of BDNF and mTORC1 signaling in rat hippocampus and PFC [5,29-31]. These two areas are the limbic brain regions directly involved in depression and antidepressant actions [32,33]. Thus, studies on the mechanism of LY341495 exerting antidepressant effects has also been investigated in PFC or mixed tissue of PFC and hippocampus. Previously, we reported an in vitro study showing that LY341495 promoted hippocampal dendritic outgrowth and synaptic protein expression via stimulating mTORC1 signaling [12]. Since it was necessary to perform in vivo study, we investigated the mechanism in the hippocampus. >

3) The proposed “mTORC1 signaling hypothesis” shown in Fig. 4 was solely based on correlation of behavioural test (FST) results with some biochemical changes. No experiments were performed in order to find out, if there is a functional link between these phenomena.

→ Thank you for your comment. Figure 4 has been revised to putative signaling observed in the hippocampus. mPFC and DRN, which were not investigated in the current study, were removed.

4) This paper needs a language revision. Some sentences are not precise enough, e.g., see the Abstract: “Therefore, we investigated the effects of chronic LY341495 treatment on signaling by mechanistic target of rapamycin complex 1 (mTORC1) and the levels of synaptic proteins in mice subjected to chronic unpredictable stress (CUS)”.

→ Thank you for your comment. The sentence described above has been corrected. The English in this document has been checked by at least two professional editors, both native speakers of English. For a certificate, please see: http://www.textcheck.com/certificate/7pZ6RE

Round 2

Reviewer 1 Report

The authors edited the text to partially answer my concerns.